# Qualitative Study of Participation Facilitators and Barriers for Emergency School Meals and Pandemic Electronic Benefits (P-EBT) in an Urban Setting during COVID-19

**DOI:** 10.3390/nu14163358

**Published:** 2022-08-16

**Authors:** Jennifer W. Cadenhead, Julia E. McCarthy, Thanh Thanh T. Nguyen, Michelle Rodriguez, Pamela A. Koch

**Affiliations:** 1The Department of Health and Behavioral Studies, Program in Nutrition, Laurie M. Tisch Center for Food, Education and Policy, Teachers College, Columbia University, New York, NY 10027, USA; 2New York Health Foundation, New York, NY 10018, USA

**Keywords:** school meals, COVID-19, parent perspectives, emergency feeding policy considerations, nutrition security

## Abstract

During the COVID-19 pandemic, significantly fewer of New York City’s (NYC’s) 1.1 million public school children participated in emergency grab-and-go meals—heightening the risk of inadequate nutrition security for many of NYC’s most vulnerable residents. This study sought to examine student families’ facilitators and barriers to participation in the grab-and-go meal service and their experiences with pandemic-electronic benefit transfer (P-EBT) funds, a cash benefit distributed when schools were closed. We recruited 126 parents of children in NYC public schools who had participated in the grab-and-go service. Using opened-ended questions, we interviewed 101 parents in 25 1-h online focus groups. We identified four main themes which broadly impacted school meal participation: communication, logistics, meal appeal, and personal circumstances. Key facilitating subthemes included clear communication, ease of accessing sites, and high variety. Key sub-themes negatively impacting participation included limited communication and low meal variety. Accurate, timely communication; easily accessible distribution locations; and convenient distribution times could have increased participation and satisfaction. For P-EBT, parents welcomed the funds and used them readily, but some experienced difficulties obtaining payments. The simultaneous inclusion of community-based research in the evaluation of emergency feeding programs could improve future outcomes for school meal participation and electronic benefits.

## 1. Introduction

In response to the challenges created by the SARS-CoV-2 infection (COVID-19) pandemic, the New York City (NYC) Department of Education (DOE) closed schools for in-person instruction on 16 March 2020 [1,2]. Despite switching to grab-and-go meals to maintain a school meal service for public school students, in the weeks following the new service’s introduction, there was a precipitous (30–60%) decrease in the number of meals being taken, compared to the period before school closures [2,3].

A significant drop in NYC school meal participation implied an increased risk to nutrition security for hundreds of thousands of vulnerable children. The United States Department of Agriculture (USDA) has recognized nutrition security as being food secure, while also having the ability to have consistent access to affordable, nutritious, and health-promoting foods, recognizing socio-environmental and behavioral concerns through a health-equity lens [4]. Prior to the pandemic, the NYC DOE’s Office of Food and Nutrition Services (OFNS) operated the largest public school meal service in the United States (U.S.). This was made possible through use of the USDA Community Eligibility Provision (CEP) to provide universal free meals to all ~1.1 million school children, as ~70% of NYC public school children were eligible for free or reduced-cost meals, to improve nutrition security [2,3,5]. Pre-COVID, OFNS served approximately 960,000 school meals daily in ~1800 NYC public schools through the National School Lunch Program (NSLP) and the School Breakfast Program (SBP) [2,3].

The SBP and NSLP have specific requirements for a healthy, diverse diet. Meals must include five components: (1) fruits, (2) vegetables, (3) whole grains, (4) dairy, and (5) meat or meat alternative, and students must select three of the five components for the federal government to reimburse the meal’s costs [6,7,8]. These nutrition requirements result in some students receiving a significant portion of their daily calories (48%), vegetables (41%), and dairy (71%) from school meals, highlighting the importance of the contribution these meals may make to nutrition security [6,7,8]. Nationally, estimates indicate that children who participate in school meals under the SBP and NSLP consume more fruits and vegetables than children who do not. School meals are among the healthiest meals that children access during the day [9] and are crucial for health and learning. Notably, NYC school meals adhere to a formal “wellness policy” that has standards that exceed those for national school meals, providing instruction beyond requiring dietary diversity of healthy foods, with, for instance, prohibitions on artificial flavors and artificial colors in beverages [10].

Especially in the early days of the pandemic, as NYC shut down, many caregivers of young children, who were already financially challenged with job loss or fear of COVID-19 infection, faced rising levels of food insecurity [11]. It was also known that food insecurity among those who were already vulnerable was rapidly increasing [12]. Changes to school meal programs and the introduction of Pandemic-Electronic Benefit Transfer (P-EBT) were designed to address this increase. USDA waivers allowed for flexible meal distribution times [13], removed the congregate meal requirement [14], and allowed parents to pick up meals [15]. OFNS responded by providing grab-and-go cold meals to students during remote instruction. Families also received P-EBT payments to alleviate the financial stress they incurred due to school closures [11,16]. This enabled families to recoup some out-of-pocket expenses for food that had previously been provided at schools.

By early March 2020, NYC had become the U.S. epicenter of the COVID-19 pandemic. Although other areas of the country would eventually have higher COVID-19 rates, by the end of March 2020, NYC had 1.5 times the number of cases and nearly nine times the number of deaths as the rest of the state combined [17]. NYC made up almost a quarter of cases and one-third of deaths nationally [17]. High COVID-19 rates and deaths coexisted in communities with higher rates of poverty [18]. These communities had high rates of underlying health conditions, such as obesity, type 2 diabetes, and cardiovascular diseases, with higher numbers of immigrants and individuals identifying as Non-Hispanic Black, Hispanic, or Latino [18]. These also tended to be in neighborhoods where, even accounting for race and underlying health conditions, people were employed in essential services—occupations requiring interaction with the community [19]. Researchers have since noted that, even with later COVID-19 variants, COVID-19 infection rates disproportionately impacted communities of color [20]. Nationally, prior to and during the COVID-19 emergency, a greater percentage of children who were eligible for SBP and NSLP from Non-Hispanic Black and Hispanic/Latino participated compared with Non-Hispanic White and other race/ethnicities [7], suggesting school meals were crucial to nutrition security for these groups.

As in other U.S. cities, despite both the need for and availability of meals, many families in NYC did not participate in the grab-and-go meal service [3]. However, the reasons for lower participation are unclear. Research prior to the pandemic has suggested that summer meal participation declines when schools are closed due to several barriers, including difficulties getting to meal settings for families with limited means [21].

There has been limited examination of the low school meal utilization during COVID-19 in urban U.S. cities from a family’s perspective [22]. Therefore, the aim of this study was to better understand emergency school meal participation facilitators and barriers for families in NYC, as well as their experience using P-EBT funds, to potentially increase nutrition security during future emergencies.

## 2. Materials and Methods

### 2.1. Research Design

We completed an observational, cross-sectional, mixed-method study that collected data from both a quantitative survey and qualitative semi-structured interviews. Here, we make available data from the qualitative portion of the study.

We recruited, surveyed, and interviewed participants in April and May 2021, with a target participation of 100 parents. We utilized purposeful sampling, recruiting participants in conjunction with community partners (i.e., Lunch 4 Learning Parents Caucus) via flyers, email, and social media (i.e., Instagram and Facebook). In addition, we placed Facebook advertisements to balance out participation among as many communities as possible and include those neighborhoods known to have high levels of socioeconomic and health challenges [23]. To be included, participants had to have been a parent or primary caregiver of an NYC public school student who had received grab-and-go meals at any point from the beginning of the COVID-19 pandemic (March 2020–May 2021), were at least 18 years old, and who were willing to participate in a one-hour English-speaking focus group with other parents about school meals. As a token of appreciation, we sent them a $25 electronic gift card for participating in the focus groups (Amazon or VISA).

A research assistant contacted potential participants who expressed interest in the study by filling out an electron interest survey, using a Google form (Google, Mountainview, CA, USA) (see Appendix A). The assistant confirmed eligibility and obtained informed consent for participation in the study, as well as for audio and video recording, by telephone. After obtaining participant consent, the research assistant scheduled participants for a video focus group (with 10 or fewer participants) and asked them to complete a Qualtrics survey form (Qualtrics, Provo, UT, USA), detailing personal information (see Appendix B). She then scheduled participants for focus groups on a separate day from the screening, including weekend and evening hours, based on their availability. Participants received reminder follow-up emails three days and one day prior to their focus group scheduled date, along with a text message one hour prior to the start. Individuals who initially expressed interest via the electronic interest survey, but were not reached after being contacted by email and/or text message on six separate days, or who failed to attend their scheduled focus group after two attempts to reschedule, were excluded. If location services were enabled on the electronic device, Qualtrics also recorded the global positioning system (GPS) location where the survey was completed. When determining whether additional recruitment to increase participation across NYC was needed, both participants’ self-reported community school district and device GPS location were taken into account.

### 2.2. Semi-Structured Interviews

We used a semi-structured interview format to conduct the recorded 45 to 60-min focus groups via video conferencing (Zoom Video Communications, San Jose, CA, USA). We reminded participants of their prior consent and assured them that they were not required to have their cameras on during the recordings. Prior to questioning, we encouraged participants to openly express their views and to allow others the benefit of expressing their views as well, without judgment. We reminded them that we, as researchers, would keep all information confidential, that reporting would be aggregated to protect individual identities, and that participants should not share any information outside the group.

We used focus group questions that had been developed for use in a separate study using a social-ecological framework among a lower-income, mostly Latino and Spanish-speaking population in California during the same time period [22] (Appendix B). Questions were determined to have face validity for use among NYC parents with a test group of parents. Questions explored the following domains: grab-and-go meal facilitators and barriers, community partnerships facilitating meal service, P-EBT, understanding meal-related needs of the family and children, future intentions, and key takeaways.

For consistency, the study coordinator (JC), who had training and experience with qualitative research and focus group interviewing, moderated all sessions. A research assistant (TN), a masters-level student in nutrition with some training in leading focus groups, was also present and made notes of questions and non-verbal communications (either through movement or chat comments). The audio recordings were transcribed automatically via the platform software. We manually corrected the transcripts generated from the audio recordings.

We initially used proposed themes that had been developed based on an understanding of concerns voiced by advocates and parents in partner meetings. After each session, the moderator and research assistant came to a consensus in identifying whether participants discussed topics aligning with proposed themes, if topics had been previously discussed in other focus groups, or if new themes had emerged. Generally speaking, multiple members of each focus group expressed common themes. A second research assistant, unfamiliar with the groups, reviewed 20% of the videos to validate findings and determine whether themes had been fully and correctly captured. Finally, we condensed themes to reflect key findings. Although data saturation appeared to have been obtained after six focus groups (31 participants), we continued interviews to more fully ensure geographic representation, and oversampled socioeconomically challenged neighborhoods to reach families with the greatest needs for the meals. We later identified one additional theme with the 14th focus group (after 58 participants), with no later additional themes in the final 11 focus groups. On average, our focus groups had four participants (range: 1–9).

We shared preliminary study results with our community partners and OFNS as quickly as possible to address potential concerns discovered.

### 2.3. Statistical Analysis

We conducted an analysis of participants who completed the initial Qualtrics survey, providing demographic and other information, as well as who had participated in a focus group. We tested for normality and the data was found to be non-parametric. Due to a lack of normality and small sample size, for categorical comparisons of two groups (gender), we used a Fisher’s exact test comparison, while we used Kruskal–Wallis equality-of-populations rank tests for three or more groups (all other categories—age, occupation, etc.). Statistical significance was set a priori at *p* < 0.05 for all determinations. We used Stata version 17 software to perform all statistical analyses (StataCorp LP, College Station, TX, USA).

## 3. Results

### 3.1. Participant Information

Demographics are reported in Table 1. In total, 175 individuals expressed interest in the study and were screened. Of these, we identified 126 (94.4% female) as eligible for the study and who participated in the personal information survey. There were 101 participants (96% female) who attended 25 1-h focus groups (79.5% participation rate from screening). We excluded 49 from focus groups for the following reasons: 20 had not picked up meals, 9 only spoke Spanish, 13 were no longer interested/inconvenient scheduling, and 7 were lost to follow-up (protocol previously stated).

Our participant demographics showed that 49.5% were under age 40 y, with <10% age 50 y+, and 40.6% were unemployed/retired/disabled or homemaking. A total of 78.2% were non-Hispanic Black or Latino/Hispanic, 73.3% had elementary school-aged children. As of July 2021, 40.6% of NYC public school students were Hispanic/Latino, 25.5% were Non-Hispanic Black, 15.1% were Non-Hispanic White, 16.2% were Asian, and 2.6% mixed/other groups. Participants represented all of the NYC Boroughs, and nearly all communities, although 54.3% were from the Bronx or Brooklyn. The DOE reports (June 2021) showed that 30% of public school students live in Brooklyn, 27% in Queens, 21% in the Bronx, 16% in Manhattan, and 6% in Staten Island [24]. As previously mentioned, our sampling intentionally targeted more communities with higher percentages of individuals who were Hispanic/Latino or non-Hispanic Black, as well as more neighborhoods in the Bronx and Brooklyn. These boroughs had a greater percentage of high-poverty schools prior to the pandemic.

### 3.2. Thematic Focus Group Findings on Participation in Grab-and-Go Meals

In terms of facilitators and barriers to participation in school meals, reported in Table 2, four themes emerged from the focus group analysis: communication, logistics, meal appeal, and personal circumstances. (Note: Participant comments below may have been edited for clarity and brevity.)

#### 3.2.1. Communication as a Facilitator to Participation in Grab-and-Go Meals

Overall, as reported in Table 3, in 96% of focus groups, parents stated that their participation in grab-and-go meals was facilitated by school communication.

Parents said there was a mix of communication types that were effective. Having multiple means of communication was particularly important as no one means reached everyone equally. In 92% of focus groups, parents said that multiple means of communication directed towards them regarding grab-and-go meals raised their awareness. Parents reported receiving numerous emails and fliers from the school or parent-teacher association (PTA), or being contacted by school staff, such as class coordinators, guidance counselors, and teachers. Some parents also mentioned school sources such as ClassDojo, a classroom app (ClassDojo, San Francisco, CA, USA), helped them to become aware of the meals. A few participants also said that teachers would make daily announcements at the beginning of the virtual class.

Parents noted external media coverage and signage as important communication tools. In 64% of focus groups, parents reported hearing about the meals, especially at the start of the pandemic, from mass media promotion, such as television announcements by the mayor, or from city-wide meetings. In 52% of focus groups, parents said informal support, such as social networks, facilitated their participation in grab-and-go meals. These parents learned about options from friends, neighbors, social media, and various ‘word-of-mouth’ methods. In 40% of focus groups, parents noted signage, in front of schools or on the meals, which encouraged their participation.

#### 3.2.2. Communication as a Barrier to Participation in Grab-and-Go Meals

Overall, as reported in Table 4, parents in 76% of focus groups said communication methods could have been improved and the lack of it was a barrier.

In 60% of focus groups, parents said that they were unclear about menu options. They reported that they were often given meals that were packaged in brown bags or thick plastic wrapping, but not labeled, and that it was difficult to discern what was in the package. Parents were concerned they would take meals that their children did not want, and food would be wasted. In addition, many said they were unable to find the expected range of options based on the menu published on the DOE website. If parents inquired about unavailable options, they were told substitutions were inevitable because the kitchen could only serve what they received due to supply chain issues.

In 48% of focus groups, parents said that there was insufficient signage. Parents wanted more real-time communication about when a site was temporarily closed, alternative options for pick up, directions for getting to sites (important for those unfamiliar with the school building layout), better signage for pick-up hours (as well as adherence to stated hours), and better notification when the site was experiencing service difficulties.

In 36% of focus groups, parents said that other parents and neighbors were unaware that food was currently available. They felt that low participation was partially a result of a lack of awareness due to inadequate marketing. Parents said that they felt communication about the meals should be more consistently shared across all school stakeholder communication (e.g., the community school districts, school-parent liaisons, principals, teachers, aides, security guards, etc.). They said that advertisements should focus on the neighborhood, not just the school population. They explained that this was necessary since children may attend a school in an area not near their home and needed to understand what options were available close to home. Parents suggested NYC DOE use multiple means to communicate about meals (e.g., postal service mail and fliers around the neighborhood), noting that internet service was not ubiquitous or consistent and that not everyone fully reads emails.

In 28% of focus groups, parents thought better two-way direct communication, in real-time, especially in multiple languages, between OFNS and families would improve participation. Parents said that they felt they were not heard and were unclear on how to express their concerns. Parents noted that signage, especially initially, but still currently, was primarily in English (with primarily English-speaking staff). Providing alternate signage in other languages would be especially useful in neighborhoods with high immigrant populations, such as Spanish in Spanish Harlem and Cantonese in Chinatown.

#### 3.2.3. Logistics as a Facilitator to Grab-and-Go Participation

Overall, as reported in Table 5, in 88% of focus groups, parents noted that the logistics of obtaining meals could be a facilitator in their participation. 

In 84% of focus groups, parents said that having a convenient location, that is, location accessibility, was an important factor in their participation. They often cited that having the location right across the street or within a five-minute walk—and close alternative locations—helped them to participate more easily. They may have had multiple small children with them or teenagers who needed to go to sites alone. They recounted needing to procure certain options at locations other than their closest location. Some parents said that being able to go to any location in the city was also useful. This flexibility allowed families to maintain privacy and avoid stigma. This also increased access for parents working away from home. A few parents noted that familiarity with their neighborhood elementary school building and personnel increased comfort and participation.

In 60% of focus groups, parents reported access to bonus resources as a participation facilitator. Parents reported being able to obtain a range of items, including books, art kits, sanitary napkins, meal kits, masks, personal protective equipment, and hand sanitizers. Even if these were offered irregularly, parents said the novelty of getting something new was a motivating factor to participate. In addition, in 52% of focus groups, parents reported that bulk pick-up was motivating for them. However, the definition of bulk pick-up could vary. For some, this meant being able to obtain breakfast and lunch meals for one day for all children in their household. For others, bulk meals meant obtaining multiple days of food at once, or a box of food with ingredients to serve a child or family for a week.

#### 3.2.4. Logistics as a Barrier to Grab-and-Go Participation

As reported in Table 6, in 80% of focus groups, parents noted that certain site logistics could be a barrier in their participation.

In 68% of focus groups, parents stated there were aspects of the location which discouraged their participation. Having to walk more than a few blocks or take long bus rides to pick-up sites was discouraging. If they needed to take the bus, parents said the cost of transit might be greater than the value of the meals. They also noted that standing in long lines in a variety of weather conditions could be a barrier. They reported limited options for seating or children playing during their extended waits. Parents said they would appreciate being able to have real-time updates on conditions (e.g., hotline or texts). They also reported that the negative impact of these issues were compounded when small children were with them.

In 64% of focus groups, parents said that limited hours made picking up meals difficult. They said that pick-up hours were initially before and after school. When pick-up hours changed from early openings at 7:30 a.m. (or possibly as late as 5 p.m.) to be more limited, generally, 9 a.m. to noon, this interfered with their children’s instructional time. These pick-up times could create conflicts with teachers expecting children to be on time, in front of cameras (without eating) during class. It also created complications managing multiple children at different grade levels or on varying schedules. Some parents who chose to continue their children in remote learning when in-person classes restarted had to wait to pick up meals until classes had started. Kitchen staff served in-person children’s meals first—up until class time—and then allowed remote children access to meals. These parents said they were told that staffing shortages and limits to their service hours meant children who were attending in-person classes needed to be served before the remote learners, even though both groups of children had to attend classes at the same time.

Parents said that other logistical issues discouraged their participation. In 48% of focus groups, parents said they were told (especially early on) that they could only pick up one meal or one day at a time. Although this process changed, this discouraged some from returning. They said that they would have liked the ability to have a box of food items for the week (or several days) to make fewer trips. In 32% of focus groups, parents said that inadequate supervision of the line lessened their willingness to participate. Parents said that, especially at the beginning of the pandemic, they felt personally unsafe due to conflicts with people living on the streets, who had behavioral, health or substance abuse issues, or may have had concerns about catching COVID in long lines or even from the food.

#### 3.2.5. Meal Appeal as a Facilitator to Grab-and-Go Participation

Overall, as reported in Table 7, in 60% of focus groups, parents said there were aspects of meal appeal that encouraged their participation.

In 36% of focus groups, parents said they were satisfied with the meal options and variety made available to them. They said they were happy with the meal rotations (e.g., weekly salads and sandwich variety). Others made comments such as that their children liked the fruit, that school meals allowed more food options than they could provide at home, and that their children looked forward to the variety. Parents who had access to hot meals said their children liked them the most. Parents who said meals at their sites adhered closely to the DOE stated menu were also more likely to report satisfaction with the meals.

In 28% of focus groups, parents said they depended on school meals because they knew they were nutritionally balanced, or their children were able to eat more fruits and/or vegetables than they otherwise would have had without school meals.

#### 3.2.6. Meal Appeal as a Barrier to Grab-and-Go Participation

Overall, as reported in Table 8, in 88% of focus groups, parents said there were aspects of meal appeal that discouraged their participation.

In 88% of focus groups, parents mentioned that menu options were repetitive and limited. For example, parents reported often being limited to the same three sandwiches: turkey and cheese, peanut butter and jelly, and American cheese daily. Similarly, they found that the options for fruit and vegetables were limited. They also rarely found salad options available. Parents had traveled to schools outside of their community and noticed inconsistencies. They wanted more uniformity across NYC, rather than feeling as if they were receiving inequitable treatment, especially in high-need communities.

In 80% of focus groups, parents reported being discouraged by the amount of food available and limited selection. Parents said they encountered instances where serving sizes were small, especially for older children and teens. Although parents said they would recycle items to incorporate into other meals, they felt that the items they were receiving were more akin to a snack, rather than a meal. Some parents said that, through word of mouth, they discovered that nearby schools had more complete menus (beyond a variety of sandwiches) than others. Parents said there were limited (and repetitive) options in most locations for children who were observing certain religious conventions (e.g., kosher, halal), maintaining vegetarian/vegan diets, had food allergies or sensitivities, or required other particular food needs, such as puree meals.

In 72% of focus groups, parents reported that there were aspects of the meals that were unappealing. Some felt the food’s presentation was unappealing or became so upon reheating. Others were frustrated due to plastic container lids for fruits or vegetables easily coming off, spilling liquid—impacting the quality and texture of items. Some said that fruit quality varied—from underripe to bruised. A few were uncertain of the food’s nutritional qualities, feeling it was ultra-processed. Some parents thought the food appeared as if it was made hurriedly, or even had questionable freshness, and wished that there could have been some more visible indication that the people making the food had indeed cared, like a smiley face sticker on the bag, or a note saying, ‘made with love,’ or ‘this too will pass’. Parents suggested having the sandwiches cut into cute shapes could have helped make them more appealing.

#### 3.2.7. Personal Circumstances as a Facilitator to Grab-and-Go Participation

Overall, as reported in Table 9, in 84% of focus groups, parents said many times, personal circumstances facilitated their participation.

In 52% of focus groups, parents mentioned informal support. Some reported having the community provide additional resources near schools which encouraged meal participation. In 44% of focus groups, parents said that the fact that the meals addressed some of their families’ basic needs facilitated their participation. The meals helped financially, particularly for those not used to budgeting for their children’s meals during the school year, increased their sense of security (especially when people were fearful to go to grocery stores), and helped maintain the normalcy associated with being in school. Lastly, the rapport that parents developed with school personnel contributed to their participation. In 36% of focus groups, parents who had developed a relationship with either the food service staff, class coordinators, or other school personnel said that the information provided by, interactions with, and gestures of these individuals helped to increase their participation.

#### 3.2.8. Personal Circumstances as Barriers to Grab-and-Go Participation

Overall, as reported in Table 10, in 88% of focus groups, parents said there were aspects of personal circumstances that acted as a barrier to their participation.

In 68% of focus groups, parents said that because they wanted to avoid food or excessive plastic packaging waste, they reduced their participation. They particularly did not wish to pick up unwanted items or have to place items in the garbage when there might be others who could utilize and be happy with the food. This appeared to correspond highly with locations where there was a limited variety of options available.

In 68% of focus groups, parents stated that family and work responsibilities were barriers to participation. Parents with young children said they could not leave their children at home, unsupervised, by themselves while parents went to pick up meals. Parents noted the requirement for their assistance during remote learning, especially for younger children, also made leaving impossible. Some had older family members who needed constant care.

In 52% of focus groups, parents stated additional personal concerns. Some had fears of contracting COVID-19. Parents reported having their own health issues which made leaving the home a challenge. Others had unpleasant staff interactions. They reported being told to be grateful they were getting anything at all—even if it was unappealing. Stigma was also an issue. Parents reported ‘swallowing pride’ and ‘doing what was necessary’. Meanwhile, their teens referred to the meal program as the ‘free-free’ and were embarrassed that they were picking up the food. Some parents traveled outside of the community school district with the hope of not having others who knew them see them; they were fearful they would be perceived as not being able to pay their rent.

### 3.3. Pandemic Electronic Benefit Payments Facilitators and Barriers to Participation

Overall, as reported in Table 11, in 80% of focus groups, parents mentioned the facilitators and benefits of using P-EBT. In addition, as reported in Table 11, in 60% of focus groups, parents discussed some barriers to using P-EBT during COVID-19.

There were a number of facilitators that increased the use of P-EBT funds. In 56% of focus groups, parents noted financial benefits. Parents reported that it helped them stretch their food budget and assisted in what they sometimes perceived to be insufficient amounts of food provided at school. They noted that P-EBT was especially helpful due to an increase in grocery store prices during the pandemic.

In 40% of focus groups, parents reported that using the benefits allowed for added flexibility. The benefits enabled parents to accommodate picky eaters with options that school meals did not allow. In 32% of focus groups, parents reported that using these cards also increased autonomy. They were able to buy groceries with the funds and cook familiar and liked foods for their family. They also reported that their children felt empowered by being able to pick out their own grocery items. In 32% of focus groups, parents reported that the benefits were easy to use. The P-EBT funds were included either on existing benefits cards, like SNAP cards, or sent on new cards that could be used in a similar way to a debit card. In 16% of focus groups, parents reported using the benefits to buy from online grocers, such as Amazon Fresh and Fresh Direct. Finally, in 12% of focus groups, families reported being able to combine with other resources, such as Health Bucks at farmers’ markets, or ‘Get the Good Stuff’ at participating grocery stores, to buy additional items. 

There were a number of barriers to using P-EBT that parents discussed. In 52% of focus groups, participants reported administrative issues. For example, parents reported that the cards never arrived, or arrived for some, but not all, children—despite repeated requests. Some parents noted that activation had been complicated, unclear, and/or difficult due to administrative errors with the date of birth of their children—which was required for activation. Parents found that school records for addresses could be inaccurate or not correctly communicated to the state which administered the benefits. Problems also ensued if child custody was shared among caregivers, especially where it was unclear who should receive the benefits. Parents who were fearful of immigration enforcement actions preferred to not use the card rather than report issues and risk other difficulties.

In 40% of focus groups, parents reported that lack of clear communication made using the benefit difficult. Some were unclear on what P-EBT could purchase (e.g., fees, online purchases, etc.), how much money was allocated, or which of their children were eligible (e.g., children attending Pre-K or charter schools).

In 24% of focus groups, parents noted that the funds were received months after their greatest need. In 16% of focus groups, parents stated that there was a lack of support from the school when they reached out for assistance with P-EBT. Parents also reported being unable to get help when they contacted the information helpline. Finally, in 8% of focus groups, parents were unable to speak specifically about P-EBT, since they were unclear when the benefits were deposited to their SNAP card, as temporary federal measures resulted in other additional benefits also being deposited on the same card.

## 4. Discussion

Given that school meals have been shown to be among the healthiest meals that children consume in the U.S. and help improve academic outcomes [9], providing school meals for children, especially during periods of emergency, is critical for nutrition security. Our results showed that government support of families with programs such as grab-and-go meals and P-EBT is welcomed, appreciated, and impactful, but could be refined to reach more families if there were more immediate community feedback built into the process. OFNS, like many school meal service programs across the nation, pivoted to a new grab-and-go emergency feeding program over one weekend at the start of COVID-19 [2]—undoubtedly, a herculean effort. This turned into a long-term feeding operation lasting at least through to the conclusion of our research (June, 2021). Given the sudden switch to a new mode of meal service operation, an evaluation with a pre-post design was not possible. However, evaluation designs that collect real-time community feedback may help during future emergencies.

Parents’ reasons for participation in school meal programs were grouped into four broad themes: communication, logistics, meal appeal, and personal circumstances. The most commonly cited facilitators for parents were: (1) communication, especially from multiple school sources, and (2) logistics, particularly having convenient pick-up locations and hours. The most commonly cited barriers fell under meal appeal, which included lack of variety and inadequate amount of food to meet their family needs. These barriers appeared to most impact participation if communications and logistics were also barriers. There were frequent discussions of the kind, *‘*it wasn’t clear my kid would like what I’d be getting; the choices were limited; would the sites be opened or closed; getting there can be difficult, would it even be worth it to go out and pick up meals?*’.* Researchers found similar barriers [21,22], however, unique to our study; comments in the focus groups suggested that these barriers compounded each other. This demonstrates how stakeholder collaboration to assure full menus are offered, clear communication, and well-organized pick-up sites may help increase uptake. Personal circumstances, such as conflicts with being able to pick up meals due to home and work responsibilities, as well as wanting to avoid food waste, also negatively impacted participation. When surveying nutrition directors nationally, the School Nutrition Association suggested some of these factors may have negatively impacted participation numbers during the fall and spring of 2020 [25]. However, this study is one of only two [22] that involved interviewing parents to understand their perspective on the potential issues influencing participation in the U.S. during the early part of the COVID-19 pandemic.

Parents said communication was an extremely important facilitator for participation in grab-and-go meals. Some schools had school personnel (i.e., principals, teachers, and staff outside of OFNS), who did a good job of promoting school meals with texts, and fliers, as well as announcements from parent coordinators, teachers, and others. However, communication was inconsistent across schools, with the main responsibility falling to OFNS. Perhaps, if the responsibility to communicate about school meals was shared by all school stakeholders—in more schools—this could have increased participation in school meals. Historically, school meals have been considered separate from the educational part of the day, since teachers do not supervise meal-times [26]. Because teachers were stretched so thin during the pandemic, and had limited prior knowledge of meal options, it is not surprising that few teachers regularly informed families of meal options. Having teachers, who interact the most, and on a daily basis with students, promote school meals could have helped raise awareness of meal availability. However, the responsibility should have been shared more broadly. Having consistent and regular promotion of school meals by other school personnel may have helped raise student and parent awareness about options and availability. Although principals had the authority to determine which meal service options were allowed in individual schools to best accommodate community needs, such as meal scheduling, halal or vegetarian options, and even hot and cold service, few parents in the focus groups knew this. Focus group participants said that consistent and constant reminders from multiple sources were needed to create and maintain awareness of the grab-and-go meals.

Communicating about, or marketing, the grab-and-go meals was a new undertaking for OFNS and NYC, as, prior to COVID-19, the OFNS meal service was automatically built into the school day. Three factors made communication challenging: (1) grab-and-go meals were a new service about which the public had no knowledge, (2) the meals required marketing to a large and diverse population, who were not always English speaking, and (3) meals were served from locations which families had no other reason to visit.

Initially, NYC issued public service messages on local television and radio stations to inform the community that grab-and-go meals were available to all school children for breakfast and lunch. However, these messages were communicated within a limited timeframe. OFNS was left to develop its own (grant-funded) new communication plan without time to test its efficacy. OFNS posted signage with information about the meals in front of many schools. However, sheltering-in-place requirements and remote learning may have meant that signage went unseen. Anecdotally, a news story published around the time of the focus groups (June 2021) mentioned a parent who had only found out that grab-and-go meals were being offered that very morning—3 months after the start of grab-and-go meals [27]. Moreover, when grab-and-go meals were first rolled out, NYC was the epicenter of the U.S. COVID-19 pandemic [17], and little was known about the transmission of COVID-19 or how to stay safe. During such chaos, messaging about a brand-new meal program may not have captured parental attention—save for those who were most in need and actively seeking aid.

Parents also reported that language barriers prevented them from understanding the meal availability and options, especially in areas with large immigrant and limited English-speaking populations. Since we only conducted focus groups in English, it is notable that this was brought up, and may be even more of an issue for the many parents who do not speak English than could be captured through this study. Nearly 40% of our population identified as Hispanic or Latino and were, presumably, aware of the concerns of their community members who were primarily non-English speaking. However, NYC has a diversity of people who speak languages other than English or Spanish, and few were represented in our sample. Despite this, inadequate diversity of language in sampling, adequate multi-lingual communication was noted as a concern. Although NYC municipal law requires that all communications are published in 10 languages [28], it is unclear whether this was happening to the extent needed. The beginning of the pandemic was also a time of increased action from the U.S. Immigration and Customs Enforcement (ICE) agency, which may have made immigrant families less likely to voice concerns, including for meals and language-appropriate signage about those meals. Helping families have more bi-directional community communication in multiple languages may have increased participation.

Parents said that logistics, particularly the ease of getting to a location and the comfort they felt in picking up meals, played a large role in determining their participation. Focus group participants who continued to regularly pick up meals (4–5 days per week), tended to live near a school distributing meals. Familiarity was also key, as families often picked up food from schools their children attended. Pick-up hours before or after school times also increased participation. Additional non-food resources, such as masks, also increased participation, possibly adding excitement at a time with few pleasures.

Conversely, transportation issues, meal availability, and challenging personal circumstances, negatively impacted participation. Parents who had to take a bus multiple stops with small children or walk more than a few (>5+) city blocks, were less likely to continue picking up meals. Waiting in long lines in difficult weather, meals repeatedly running out, school pick-up sites closing due to COVID-19 quarantine requirements, and lack of food labeling were also important factors impacting participation. Parents reported that personal circumstances, such as having other family members with health challenges, conflicting work schedules, not wanting to waste food, and feeling stigmatized, unsafe, or unwelcomed when picking up the meals either stopped, or significantly curtailed, participation.

Parents had a mixture of opinions about the appeal of the meals. Although many parents expressed concerns, these appeared most acutely when parents encountered a limited variety or limited amounts of food. Others cited inadequate quantity or unappealing preparation. However, there were parents who felt their children received adequately prepared meals of a good variety. In addition to providing financial relief and normalcy, some parents noted that their children were more likely to consume milk, juice, fruits, and vegetables from school meals. Concerns about meal appeal were most likely to negatively impact participation when coupled with other barriers, described above. Parents who obtained meals at locations that offered a variety of meals adhering to the choices stated on the DOE menu, such as salads and wraps, were more likely to report meal acceptance and appeal. Even with supply chain issues, parents wanted to feel they were being offered reasonable choices and were heard and respected. Low variety and inconsistencies in meal availability across schools left some parents angered and frustrated. While OFNS faced similar challenges to those widely reported in food service, including food supply chain issues, high levels of demand, and having staff unable to work due to having COVID or fear of COVID exposure, parents did not feel they were getting empathetic responses when they failed to receive the variety of options stated on the menus. Being able to maintain high fidelity to the stated menu may have created more support and higher meal participation.

The School Nutrition Association has also noted that COVID-19 exacerbated school meal staff shortages [25]. They noted that this was due to a tight labor market and being unable to replace workers who left due to illness, early retirement, to care for family members, fear of COVID-19 exposure, and exhaustion, as well as supply chain challenges. Being significantly short-staffed, including in supervisory roles, likely negatively impacted program fidelity. Given the stated importance of school meals to participants’ families, the data presented here can hopefully help inform school meal authorities, and secure commitments from legislators to provide more comprehensive support for school food operations during future school closures, including for summer service, scheduled holidays, and emergencies.

Our data collected on P-EBT may also inform future feeding programs for children. As with school meals, parents reported being grateful for the P-EBT benefits. They used words like ‘lifesaver’ and ‘much needed.’ For many, using these benefits was straightforward. The majority of the families appeared to have received the benefits and had few issues with using them.

However, the issues parents encountered with the program were not always easy to resolve, with parents in 60% of focus groups reporting assorted concerns. For example, parents in up to 40% of focus groups reported issues such as lost cards, misunderstandings on which benefits card (Supplemental Nutrition Assistance Program or Medicaid) they should use to access P-EBT benefits, without clarity on where or how to get help, or not knowing when P-EBT was loaded onto their card. This confusion may be partially explained by a significant time lag between school closures—the date benefits accrued—and benefit distribution. For example, benefits accrued for spring 2021 closures, yet only began being distributed in June 2022, over a year after parents had to pay for missed school meals. Some parents were unaware of, or confused about, the additional benefits because temporary SNAP payments arrived on the same card, without clarifying communication.

To alleviate confusion, parents reported that New York State published a website with additional information and fielded a hotline. However, few parents reported being able to get specific questions answered or problems solved for P-EBT. Importantly, estimating the impact of P-EBT was difficult due to the lack of clarity for these participants on whether they had even received the benefits. Going forward, for programs of such economic magnitude, funding for clear communication upon roll-out and more live assistance are needed.

This study has strengths and limitations. None of the authors have experienced a lack of nutrition security, nor did they have children in the NYC public school system at the time of this study. Therefore, there may be nuances of the lived experience which may not have been plainly identified with these perspectives. However, several of the authors are working parents and have had children who attended public school. In addition, the authors have spent years studying school meals and nutrition security issues, including those in NYC, and the authors have diverse ethnic identities and social experiences. Therefore, this cumulative experience may represent a study strength.

Fewer parents employed outside of the home, as well as fewer families with the youngest categories of children, were able to participate in the focus groups, compared with those who initially were eligible to participate, which may have been a limitation. Language was another limitation, as previously discussed. However, the percentage of focus group participants who identified as Hispanic/Latino was similar to DOE statistics.

The views expressed by the sample may also be a limiting factor—particularly if the pool of participants in the sample was not representative of the actual population. Qualitative research does not develop a set hypothesis to test against a null hypothesis. Rather, it seeks to understand experiences more holistically and, as in this case, provide an initial examination of the phenomenon. Therefore, it does not generally involve a priori sample size determinations. We collected our data using purposeful sampling and obtained representation from all boroughs and nearly every community school district, oversampling communities that were presumed to have a higher need for grab-and-go meals. Nonetheless, a larger, statistically representative sample may have revealed additional or different findings. However, with the exception of one topic, relating to immigration, we achieved theme saturation after a limited number of focus groups—with additional focus groups only confirming earlier findings. Therefore, we feel that this repetition of findings added to the reliability and validity of the findings and was a strength. To our knowledge, this is the only study of a large urban school district from a family perspective of school meal distribution during the COVD-19 pandemic in the U.S. Studying the impact of new programs concurrent with their operation was a strength.

## 5. Conclusions

The grab-and-go meal service and P-EBT reached a large number of NYC public school students in the initial portion of the COVID-19 pandemic and may have improved nutrition security and contributed to a healthier meal pattern, normalcy, and financial security for participating families. However, more students may have participated in grab-and-go meals if there had been greater awareness of the program and fewer barriers to accessing meals. More consistent variety of meal options, accurate, timely communication, easily accessible locations, and convenient distribution times could have increased participation and satisfaction. Greater access to live helplines may have alleviated confusion in using P-EBT funds. During future school emergencies, collecting and incorporating community feedback could result in greater participation and satisfaction, as well as in increase in nutrition security for children and their families.

## Figures and Tables

**Table 1 nutrients-14-03358-t001:** Demographic information comparing survey-only versus focus-group-completing participants.

	Survey (*n* = 126)	Focus Groups (*n* = 101)	*p*-Value
**Gender** Female	119 (94.4%)	97 (96.0%)	0.014
**Age** <35 35–39 40–44 45–49 50+ Unknown	34 (27.0%) 32 (25.4%) 33 (26.2%) 15 (11.9%) 11 (8.7%) 1 (0.8%)	24 (23.8%) 26 (25.7%) 27 (26.7%) 14 (13.9%) 9 (8.9%) 1 (1.0%)	0.109
**Occupation** Unemployed/ Retired/Disabled Homemaking Office/White-Collar Education Blue-collar Healthcare Government Other/Unknown	24 (19.0%) 21 (16.7%) 21 (16.7%) 15 (11.9%) 12 (9.5%) 9 (7.1%) 4 (3.2%) 20 (15.9%)	22 (21.8%) 19 (18.8%) 16 (15.8%) 12 (11.9%) 8 (7.9%) 8 (7.9%) 3 (3.0%) 13 (12.9%)	0.007
**Race/Ethnicity** Hispanic/Latino Non-Hispanic Black Non-Hispanic White Asian Mixed/Other/Unknown	52 (41.3%) 44 (34.9%) 14 (11.1%) 10 (7.9%) 6 (4.8%)	40 (39.6%) 39 (38.6%) 10 (9.9%) 9 (8.9%) 3 (3.0%)	0.503
**Home language** English Spanish Other	106 (86.2%) 12 (9.8%) 5 (4.0%)	86 (88.7%) 9 (9.3%) 3 (3.0%)	0.449
**Borough/Community school district of the youngest child** Brooklyn The Bronx Manhattan Queens Staten island Special Education ^1^ Unknown	38 (29.9%) 31 (24.4%) 23 (18.1%) 17 (13.4%) 6 (4.7%) 4 (3.1%) 8 (6.3%)	33 (32.7%) 25 (24.8%) 18 (17.8%) 16 (15.8%) 3 (3.0%) 4 (4.0%) 2 (2.0%)	0.602
**Families with children in age group** ^2^ Younger than Pre-K Pre-K–5th grade 6th–8th grade 9th–12th grade Beyond high school	19 (15.2%) 94 (75.2%) 39 (31.2%) 44 (35.2%) 11 (8.8%)	11 (10.9%) 74 (73.3%) 33 (32.7%) 37 (36.6%) 10 (9.9%)	0.008

^1^ The Special Education district draws from all five NYC Boroughs. ^2^ Statistical differences are based on the age of the youngest child in the family. Statistical significance set at *p* < 0.05.

**Table 2 nutrients-14-03358-t002:** Themes for participation in grab-and-go meals during COVID-19.

Themes:	Communication	Logistics	Meal Appeal	Personal Circumstances
	Facilitating Communication Subthemes*:*	Facilitating Logistics Subthemes:	Facilitating Meal Appeal Subthemes*:*	Facilitating Personal Circumstances Subthemes:
Facilitators	Multiple Methods of School Communications Mass Media Promotion School & Meal Signage	Location Accessibility Bonus Resources Bulk Pick-up	Meal Options Nutritional Quality	Informal Support Basic Needs Support School Personnel Rapport
	Barrier Communication Subthemes*:*	Barrier Logistics Subthemes*:*	Barrier Meal Appeal Subthemes*:*	Barrier Personal Circumstances Subthemes*:*
Barriers	Unclear Menu Choices Inadequate Signage Inadequate Marketing Inadequate Two-Way Communication	Location Limitations Limited Hours Limited Bulk Pick-up Lack of Supervision	Limited Variety & Inconsistent Options Limited Selections for Special Diets Unappealing Meals	Food & Excess Packaging Waste Avoidance Family & Work Responsibilities Miscellaneous Challenges

**Table 3 nutrients-14-03358-t003:** Communication facilitators to grab-and-go meal participation during COVID-19.

Communication	Subtheme Topics	Focus Groups	Selected Quotes
Facilitating Communication Subthemes	Multiple methods of school communication	92%	“My school made a flyer and a PC [parent coordinator], sent it out numerous times to come pick up the grab-in go. And then she also explained, you know, the timing of the grab and go at the PTA meeting. So yeah, they were good at giving out the news…”
	Mass media promotion	64%	“[Mayor] DeBlasio said it one time and then that’s when I started getting it. I heard it on the news.” ”I would say the advertisements on television. I would see meals for children advertisements online as well.”
	Informal Support	52%	“Well, I’m a parent coordinator, and, uh, I speak to the other parent coordinators from the schools in our district… we kind of help each other out, because it’s the time for our families to be able to help everyone out not just our families in our school, it’s our community. So, like, our school, if we’re having a food pantry day, we’ll advertise it for our families, and we’ll advertise it for the community, but we also share it with the other schools so that they can send their families over as well. And vice versa.”
	School & meal signage	40%	“They put the sign up outside [of the school]” “The signage on the school. People walk by all the time, so I think that is important. Especially with the pandemic, a-lot of people weren’t out, so if there is a big banner outside the school, people tend to take notice.”
Number of groups mentioning communication as a facilitator	24 out of 25 focus groups	96%	

**Table 4 nutrients-14-03358-t004:** Communication barriers to grab-and-go meal participation during COVID-19.

Communication	Subtheme Topics	Focus Groups	Selected Quotes
Barriers Communication Subthemes	Unclear menu choices	60%	“I had to question [the available options], ‘Well, that’s not what ---that wasn’t on the menu?’ And then I had to hear, ‘Well, that’s what they gave us today…[or]that’s what—that’s what’s left.’”
	Inadequate signage/ communication	48%	“There’s no signage on the school. I only know that I can go there to pick up the food when we did pick it up was because I Google everything. I look. I search. I make sure to find out where the, the information is. So, from my end, our school is … a total lack of communication. There’s no signage on the doors. There’s no message from the school leadership like, ‘hey, the school’s closing…, you can pick up at this time…, you can pick up breakfast…, if you need extra.’ There’s just absolutely crickets on it from our school.”
	Inadequate marketing	36%	“From my school, there’s been a lack of communication from the leadership in the school as per what time we can pick up the food. Like what the other parents said, if it closes, where can you get it?”
	Inadequate two-way communication	28%	”They don’t give you no information… I Googled, [my child’s school] giving meals.’ And, it pops up. Yes. Then I went onto the Board of Ed, and I saw the listing there... But, they just didn’t say nothing to us.” “… one of my schools was one of the first ones have signs up. But again, it was only in English. So, the neighborhood is a mix of, you know, Spanish, Asian. There are Caucasians. But the language was only… only one language. So, you know, the neighborhood is very diverse. You need to have it in multi language”s.
Number of groups mentioning communication as a barrier	19 out of 25 focus groups	76%	

**Table 5 nutrients-14-03358-t005:** Logistics facilitators to grab-and-go meal participation during COVID-19.

Logistics	Subtheme Topics	Focus Groups	Selected Quotes
Facilitating Logistics Subthemes	Location Accessibility	84%	“It’s within walking distance of the apartment.” “It’s easy because it’s just right down the block.” “It’s easier for [my daughter] and more convenient to just go across the street, go to the school, pick up a lunch, and, and eat it, instead of also waiting for me to come home and cook for her.”
	Bonus Resources	60%	“They gave out resources … anytime you went and got a meal, they usually had a paper that they handed you that told you about other resources. They also gave out masks at one time.” ”I know, some schools, some places that were also giving feminine products as well. Yeah, with a lot, you know, you could pick up feminine products. You could, like she said [get] the masks, some gloves.”
	Bulk Pickup	52%	“[It’s easy for me since] I have three kids, and we’re picking up breakfast- because … they give us breakfast and lunch at the same time, which I love.”
Number of groups mentioning logistics as a facilitator	22 out of 25 focus groups	88%	

**Table 6 nutrients-14-03358-t006:** Logistic barriers to grab-and-go meal participation during COVID-19.

Logistics	Subtheme Topics	Focus Groups	Selected Quotes
Barriers Logistics Subthemes	Location Limitations	68%	“I’m not walking 7 blocks just to get peanut butter & jelly when I can make it at home.” “The school that was offering food was close to 10 blocks away from me or so.” [Meant to imply she was not going to travel that far for meals.]
	Limited Hours	64%	On the challenge of getting the meals right before class starts for multiple children: “I have 10 kids. …once we have the grab-and-go, I just give everybody their breakfast… and they eat while they’re on the tablet. …some of the teachers would be like, you know, “Don’t eat!” Or, “Stop eating!” But, they don’t understand, like, what’s going on in your house—it’s so hectic to serve a breakfast, you know -- on the breakfast table—when we have school at 8:30 a.m. That is like, really hard!”
	Limited Bulk Pickup	48%	“Well, I think that having it [bulk] delivered would, would help a lot.” “Um…actually, I didn’t have a good experience with getting food because I do have a large family. I have a household of seven. Umm…so they would limit me to only taking three meals, sometimes…just 3 meals for 3 people.”
	Lack of Supervision	32%	“I felt like it wasn’t supervised. And, you know, it was a little chaotic. And the second time I went to pick up, and it was just like, not safe, or I didn’t feel safe, because the other people were too close. …my child was like, ‘Oh, I don’t want to stand here. It’s too many people.’”
Number of groups mentioning logistics as a barrier	22 out of 25 focus groups	80%	

**Table 7 nutrients-14-03358-t007:** Meal appeal facilitators to grab-and-go meal participation during COVID-19.

Meal Appeal	Subtheme Topics	Focus Groups	Selected Quotes
Facilitating: Meal Appeal Subthemes	Types of options available	36%	“I really loved the breakfast and milk and fruit and veggies.” “My kids’ picky, but he eats it.” “When they started doing hot lunches, that helped out a lot too, because it was more of a lot of variety. And then, it was like, the lunch, the fruit, the snack, which was great.” “As much as there really wasn’t a large selection of things, that probably would have been the same things that I would have packed for [my children] and sent them with to school.”
	Nutritional quality	28%	“My child eats a lot more vegetables because of the school food.” “They were fresh. I heard some complaints that at some places they were stale, but they were fresh—they were accommodating, and it was pretty well balanced. There was the fruit, there was juice, there was milk, and there was also a snack…like a granola bar or like a pack of apples. So, I think…that was pretty good.”
Number of groups mentioning meal appeal as a facilitator	15 out of 25 focus groups	60%	

**Table 8 nutrients-14-03358-t008:** Meal appeal barriers to grab-and-go meal participation during COVID-19.

Meal Appeal	Subtheme Topics	Focus Groups	Selected Quotes
Barriers Meal Appeal Subthemes	Limited Variety & Inconsistent Options	88%	“We’ll strictly get sandwiches. We get nothing else. It’s either peanut butter, cheese- strict American cheese on whole wheat bread and that’s it…” “I’m seeing pictures of other schools’ [grab-and-go] lunch, it was nothing compared with what I saw here in front of my building. It was like,’ Why are we not having the same thing?’ ‘What’s going on here?’ I don’t want to play the discrimination card, but it was like what’s going on?”
	Insufficient Food Items & Limited Selections for Special Diets	80%	From a parent needing kosher options, with a teenage son: “I was very disappointed because it was always like hummus, crackers, and… I was just like, ‘this is not a meal, it’s more like snacks to me.”
	Unappealing Meals	72%	“The containers that these vegetables come in, they are not well sealed. So, then they will spill onto whatever fruits [that’s in] there, the sandwiches … and then [my children] just don’t want to eat it. … then the cheese melts into the turkey, and mine don’t eat cheese, so I’m there peeling off, trying to scrape it off and then the bread doesn’t work so it’s just um, the lack of options.” “It looked like, ‘be thankful we’re giving you something to eat, you know? We’re in the middle of a pandemic, you should be grateful!’ That was the message I got from it.”
Number of groups mentioning meal appeal as a barrier	22 out of 25 focus groups	88%	

**Table 9 nutrients-14-03358-t009:** Personal circumstance facilitators to grab-and-go meal participation during COVID-19.

Personal Circumstances	Subtheme Topics	Focus Groups	Selected Quotes
Facilitating Personal Circumstances Subthemes	Informal Support	52%	“I got lucky. My son’s high school (school in Brooklyn)—actually when the pandemic started, they gave us all vouchers to have food delivered directly to our house.”
	Addresses Basic Needs	44%	On the fact that her middle school sons were constantly hungry and needing the meals: “If the house was made of gingerbread, I probably wouldn’t have a house.” ”[Grab-and-go meals] provided some sort of normalcy, if you will, for, especially for my older one because it was a school lunch, right, it’s what she was accustomed to seeing during lunch time. I was gonna get milk. I was gonna get a fruit. I was gonna get something else to eat and vegetable that I wasn’t going to touch. So for her at the very beginning, it helped a little bit with ‘Okay, I have class, I have lunch, I have class.”
	School Personnel Rapport	36%	On a privileged friendship at the school, “I have a friend there. And I say, ‘Send me a text’…, I say, ‘Y’all have salad today?’ And then when I’m coming back from washing, I’ll pick it up.”
Number of groups mentioning personal circumstances as a facilitator	21 out of 25 focus groups	84%	

**Table 10 nutrients-14-03358-t010:** Personal circumstances barriers to grab-and-go meal participation during COVID-19.

Personal Circumstances	Subtheme Topics	Focus Groups	Selected Quotes
Barriers Personal Circumstances Subthemes	Food & Excess Packaging Waste Avoidance	68%	“An overkill of packaging.” “Literally, it was like the big gallon plastic bag filled with food... and this is a very pervasive issue in our society, in our city.”
	Family & Work Responsibilities	68%	“Since my child has ASD, it was very hard at the beginning to take him out, just because it was hard for him to even wear a face covering.” “It was hard, because sometimes our kids, sometimes have disability that may not allow them to be able to go outside, because they might be in a wheelchair or different things.
	Miscellaneous Challenges	52%	“I mean, the whole point of COVID is, you know, to stay remote and not to be outside in public and you’re making us go every day. It kind of defeats the purpose.” “I’m of on the sixth floor. I got asthma, COPD, diabetes. I got a lot of health issues. I cannot be going out much every single day, you know?” On navigating getting the meals if your child is in a wheelchair: “It was hard.”
Number of groups mentioning personal circumstances as a barrier	22 out of 25 focus groups	88%	

**Table 11 nutrients-14-03358-t011:** P-EBT facilitators and barriers of usage during COVID-19.

P-EBT Themes	Subtheme Topics	Focus Groups	Selected Quotes
P-EBT Facilitators	Helpful funding	56%	On helpfulness: “Yeah, … that P-EBT thing did help, because I have a child that… [has] a feeding tube that can’t eat normal food, so they require baby food.” On increased autonomy: “That P-EBT card helps a lot because … At least you’re able to control what you’re buying… When you were able to find the sale… you were able to get the fresh fruits. … you were able to get the fresh vegetable.” On the relief of getting the funds when times were grim, “For me, it was extremely beneficial because my husband was the only one qualified for unemployment. …. when it hit, I was like … ‘Where did this money come from? [extremely animated] God? [laughing] It was really… it was like a weight lifted. It was a relief because it was at a time, where we were all kinds of panicking.”
	Allows flexibility	40%	
	Easy to use	32%	
	Increased autonomy	32%	
	Ability to purchase food online	16%	
	Combination with other resources	12%	
Number of groups mentioning facilitators for using P-EBT	20 out of 25 focus groups	80%	
P-EBT Barriers	Administrative issues	52%	On confusion over not receiving benefits: “I haven’t gotten mine. But I did email, um, I did email, and there’s, they’re gonna give it. They’re gonna send it in the month. I just don’t know how much is the amount? Okay? But, I did get in contact with somebody. They, say [that] they put it in a card that was [from] 2016. They just said they were gonna send me another one. And that was like two weeks ago.”
	Lack of information	40%	
	Late arrival	24%	
	Lack of support	16%	
	Confusion of P-EBT with SNAP	8%	
Number of groups mentioning barriers for using P-EBT	15 out of 25 focus groups	60%	

## Data Availability

The data presented in this study may be available on reasonable request from the corresponding author. The data are not publicly available due to confidentiality concerns.

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
