# Peer review of "Qualitative Study of Participation Facilitators and Barriers for Emergency School Meals and Pandemic Electronic Benefits (P-EBT) in an Urban Setting during COVID-19"

_nutrients, 2022, doi:10.3390/nu14163358_

Round 1
Reviewer 1 Report
The study titled “Qualitative Study of Parent Perspectives of Emergency Feeding 2 in an Urban Setting During COVID-19” was reviewed.
The manuscript was well written except for some minor sections. Abbreviation usages should be checked. The writing style of the results may be reorganized. The authors should discuss their tables with more up-to-date sources and in a more critical and rigorous manner. The main limitation is their sample size. However, they also have critical strengths.
I think that the overall manuscript is sound for publication after some revisions. Please find some suggestions in the attached file.

Author Response
Dear Reviewer 1:
Thank you for taking the time to provide us with your considered feedback. We appreciate it. We have attempted to make changes that we hope will be satisfactory to you, also taking into account comments from the other reviewer. We believe your comments have made the paper stronger.
Sincerely,
Cadenhead, et al.
Introduction
Line 87 Abbreviate New York City as NYC
Thank you for pointing this out. It has been abbreviated as NYC.
Line 88: for reference [25, 26, 27] use [25-27] instead
Thank you for bringing this to our attention. Any instances which contain three or more consecutive numbers have been hyphenated.
Line 90: Please add your aim of the study more clearly
Thank you for letting us know it was not phrased clearly.
Instead of: “Understanding the phenomenon of unexpectedly low participation from a family perspective could help improve program utilization during future unforeseen school closures, as well as reduce nutrition and food insecurity.”
We wrote in the last sentence of the introduction section,
"Therefore, the aim of this study was to better understand emergency school meal participation facilitators and barriers for families in NYC, as well as their experience using P-EBT funds, to potentially increase nutrition security during future emergencies.”
Line 104: Change grab-n-go to grab-and-go
Thank you for pointing out this to typographical error. It was been corrected.
Line 109: It can be deleted “(TN)”
Thank you. The abbreviation was deleted.
Author’s Positionality
Line 164-198: The authors gave a lot of details here. I don't think that much detail is needed.
Thank you for your suggestion. We included so much because we initially felt that in performing qualitative research, understanding the lived experience of the researcher and their positionality could inform potential bias in interpreting themes. However, since the second reviewer pointed out there was no requirement for this section in this journal and s/he also felt we should eliminate the whole section, we decided to remove the section. We felt that since a short biography will be included, we could make a statement on our potential for bias in the discussion section instead. It reads:
“None of the authors have experienced a lack of nutrition security, nor did they have children in the NYC public school system at the time of this study. Therefore, there may be nuances of the lived experience which may not have been plainly identified with these perspectives. However, several of the authors are working parents and have had children who attended public school. In addition, the authors have spent years studying school meals and nutrition security issues, including those in NYC, and the authors have diverse ethnic identities and social experiences. Therefore, this cumulative experience may be a study strength.”
Results
LINE 200: please add statistical analysis as a subheading and give further explanation about it.
Thank you for pointing out this important omission. We have now included this section. It reads:
2.3 Statistical Analysis
We conducted an analysis of participants who completed the initial Qualtrics survey, with demographic and other information, as well as participated in a focus group. We tested for normality and the data was found to be non-parametric. Due to a lack of normality and small sample size, for categorical comparisons of two groups (gender), we used a Fisher’s exact test comparison, while we used Kruskal-Wallis equality-of-populations rank tests for three or more groups (all other categories – age, occupation, etc.). Statistical significance was set a priori at p<0.05 for all determinations. We used Stata version 17 software to perform all statistical analyses (StataCorp LP, College Station, TX, USA).
Line 221: Some p-values are missing
Thank you for pointing this out. Per the combined suggestions of the reviewers, we have included all p-values as values, highlighting those which were significant.
Line 235: Referring to Table 3: please give the numbers' explanations as footnotes.
Thank you for your suggestion. We decided that it was confusing to include the number of the focus groups where the sub-themes appeared. In addition, since there was not a lot of additional value besides knowing that certain sub-themes appeared in a percent of focus groups, we decided to just delete the numbers and leave the percent of groups where comments appeared only.
Line 249: Referring to 52%: half of the groups? not most
Thank you for pointing this out, we felt that with so many different numbers -- as you correctly pointed out -- it could potentially be confusing and/or repetitive. Therefore, we decided to let the numbers stand-alone and removed all descriptive words like most, nearly half, etc. preceding the numbers. We will only show the numbers without any type of descriptive word for the percent.
Line 258: Referring to Table 4: please give the numbers' explanations as footnotes.
Thank you. Please see the comment for line 235.
Line 259: Referring to “In most focus groups (60%)”: The authors sometimes may change this terms. They seem to use the term "most focus groups" all the time
Thank you. Please see the comment for line 249.
Line 296: In referring to three highlighted paragraphs following Table 5: please give the table firstly and then make its explanation. Please check entire manuscript.
Thank you for bringing this to our attention. We mistakenly thought that the table should be inserted after the first mention to save space, but realized the actual table should be placed after the detailed description. All tables have been placed after the descriptive text throughout the manuscript. The text was also condensed slightly, where possible. However, we also note that in the MDPI style guide they state that there may be editorial changes that may rearrange the ordering of appearance of tables and corresponding descriptive text.
Line 320: In referring to Table 6: what about other quotes?
Thank you. We have added additional quotes. We also added more quotes anywhere in the text where there was only one line of quotes.
Line 321: 68% is highlighted (assume this reference is to using ‘most’ prior to number)
Thank you. Please see the comment for line 249.
Line 359: Selected quotes is highlighted (we interpreted this as you meant you would like us to add more quotes for the nutritional quality topic)
Thank you. We have added additional quotes.
Line 647: Instead of COVD-19 it should be COVID-19
Thank you for catching this typographical error. It has been corrected
Reviewer 2 Report
Manuscript of Cadenhead et al. presents an interesting insight into opportunity of grab-and-go meals during the COVID-19 pandemic and reasons for decreased participation in the program during COVID-19. However, some changes must be made to improve the quality of manuscript, namely:
Title
-I would suggest to change the title to be more related to covered topic because in current form it may be a bit misleading for reader.
Abstract
-The aim of the study must be clearly indicated.
Introduction
-Line 36 – do you have any data how many children are at risk of food insecurity?
-Lines 45 and 46 – could you explain what does it mean that schools meals exceed national standards for meal? Please provide a brief characteristics of served school meals.
-After reading the whole Introduction one issue is ambiguous – at first, Authors write that after closure of schools due to COVID-19 school meal service was switched to grab-and-go meals. However, the actual aim of the study is to examine the reasons for lower-than-expected participation in the grab-and-go meal service after closure of the schools. Please clarify it.
-Line 68 – remove double space.
-Line 78 – remove double space.
-The aim of the study must be clearly indicated at the end of this paragraph.
Materials and Methods
-Did you perform sample size calculation? As written in the Introduction, approximately 1.1 million school children participated in emergency grab-and-go meals, the final study sample of 100 parents seems to be rather small.
-What is the purpose of presenting Authors’ positionality? According to Nutrients’ guidelines there is no need to present this section, as it is not associated with the research. Please remove it.
-Information regarding statistical analysis is completely missing. How did you calculate p-value to compare survey and focus groups? It should be clarified.
Results
-Please complete specific p-values in Table 1.
-While describing the Tables and headings for the Tables it should be marked that it was during the COVID-19. Otherwise it may be unclear for the readers.
-The description of each tables should be placed above the table, not below.
-Please expand the headings for Table 3, 4, 5, 6, 7, 8, 9, 10, 11 because right now they are a bit vague.
-Line 372 – change the size and font according to the journal’s guidelines, as well as font size in Table 8.
Discussion
-In lines 504-505 Authors must add an appropriate reference to support cited information.
Author Response
Dear Reviewer 2:
Thank you for taking the time to provide us with your considered feedback. We appreciate it. We have attempted to make changes that we hope will be satisfactory to you, also taking into account comments from the other reviewer. We believe your comments have made the paper stronger.
Sincerely,
Cadenhead, et al.
Title
1.) I would suggest to change the title to be more related to covered topic because in current form it may be a bit misleading for reader.
Thank you for pointing out the potential to mislead readers.
Instead of “Qualitative Study of Parent Perspectives of Emergency Feeding in an Urban Setting During COVID-19”
We have changed the title to “Qualitative Study of Participation Facilitators and Barriers for Emergency School Meals and Pandemic Electronic Benefits (P-EBT) in an Urban Setting During COVID-19”
Abstract
2.) The aim of the study must be clearly indicated.
Thank you for this comment. We changed the second sentence of the abstract:
Instead of:
“Limited research has examined the reasons for lower-than-expected participation in the grab-and-go meal service or students’ families' experiences with Pandemic-Electronic Benefit Transfer (P-EBT) funds, a cash benefit distributed when schools were closed.”
It reads:
“This study aims to examine students’ families’ facilitators and barriers to participation in the grab-and-go meal service and their experiences with Pandemic-Electronic Benefit Transfer (P-EBT) funds, a cash benefit distributed when schools were closed.”
Introduction
3.) Line 36 – do you have any data how many children are at risk of food insecurity?
We added the bolded words after the next line to reflect the number who are at risk for food insecurity:
“Prior to the pandemic, the NYC DOE’s Office of Food and Nutrition Services (OFNS) operated the largest public school meal service in the United States (U.S.). This was made possible through the use of the USDA Community Eligibility Provision (CEP) to provide universal free meals to all ~1.1. million school children, as ~70% of NYC public school children were eligible for free or reduced meals, to improve nutrition security.”
4.) Lines 45 and 46 – could you explain what does it mean that schools meals exceed national standards for meal? Please provide a brief characteristics of served school meals.
Thank you for pointing out that this is unclear. We added language in two places in this paragraph to further clarify, which is bolded and underlined:
The SBP and NSLP have specific requirements for a healthy, diverse diet. Meals must include 5 components: 1) fruits, 2) vegetables, 3) whole grains, 4) dairy, and 5) meat or meat alternative, and students must select 3 of the 5 components for the federal government to reimburse the meal’s costs [6-8]. These nutrition requirements result in some students receiving a significant portion of their daily calories (48%), vegetables (41%), and dairy (71%) from school meals, highlighting the importance of these meals may make to nutrition security [6-8]. Nationally, estimates indicate that children who participate in school meals under the SBP and NSLP consume more fruits and vegetables than children who do not. School meals are among the healthiest meals that children access during the day [9] and are crucial for health and learning. Notably, NYC school meals adhere to a Wellness Policy that has standards that exceed those for national school meals providing instruction beyond requiring dietary diversity of healthy foods, with, for instance, there are prohibitions on artificial flavors and artificial colors in beverages [10].
5.) After reading the whole Introduction one issue is ambiguous – at first, Authors write that after closure of schools due to COVID-19 school meal service was switched to grab-and-go meals. However, the actual aim of the study is to examine the reasons for lower-than-expected participation in the grab-and-go meal service after closure of the schools. Please clarify it.
Thank you for letting us know it was not phrased clearly. We modified the last sentence of the introduction to clearly state the study’s aim:
Instead of: “Understanding the phenomenon of unexpectedly low participation from a family perspective could help improve program utilization during future unforeseen school closures, as well as reduce nutrition and food insecurity.”
We wrote in the last line of the introduction,
“Therefore, the aim of this study was to better understand emergency school meal participation facilitators and barriers for families in NYC, as well as their experience using P-EBT funds, to potentially increase nutrition security during future emergencies.”
6.) Line 68 – remove double space.
Thank you for pointing out this typographical error. Unfortunately, we could not find it. We hope that upon final edits with the manuscript, should it be accepted, it will be removed.
7.) Line 78 – remove double space.
Thank you for pointing out this typographical error. It was removed.
8.) The aim of the study must be clearly indicated at the end of this paragraph.
Thank you. Please see the response to comment #5 above.
Materials and Methods
9.) Did you perform sample size calculation? As written in the Introduction, approximately 1.1 million school children participated in emergency grab-and-go meals, the final study sample of 100 parents seems to be rather small.
Thank you for your comment. We did not perform any sample size calculations. However, we felt that given the nature of qualitative data analysis, a sample of 100, was adequate to gain a working explanatory hypothesis should a representative sampling be undertaken in the future-- since no examination had been performed to understand the family perspectives in NYC. The fact that data saturation, that is, the point where people repeatedly relayed the same major themes and sub-themes in different focus groups, was achieved early in our process of data collection suggested to us that our data was reliable and valid. However, we do acknowledge that our sampling methods may have biased our results. So, we tried to additionally clarify the strengths and limitations of this approach in the discussion section.
We wrote:
“The views expressed by the sample may also be a limiting factor-- particularly if the pool of participants in the sample was not representative of the actual population. Qualitative research does not develop a set hypothesis to test a null hypothesis. Rather, it seeks to understand experiences more holistically and, as in this case, provide an initial examination of the phenomenon. Therefore, it does not generally involve sample size determinations a priori. We collected our data using purposeful sampling and obtained representation from all boroughs and nearly every community school district, oversampling communities that were presumed to have a higher need for grab-and-go meals. Still, a larger, statistically representative sampling may have revealed additional or different findings. However, with the exception of one topic, relating to immigration, we achieved theme saturation after a limited number of focus groups – with additional focus groups only confirming earlier findings. Therefore, we felt this repetition of findings added to the reliability and validity of the findings and was a strength.”
10.) What is the purpose of presenting Authors’ positionality? According to Nutrients’ guidelines there is no need to present this section, as it is not associated with the research. Please remove it.
We felt that in interpreting the results of qualitative research, understanding the lived experience of the researchers, aka the authors’ positionality, which is sometimes disclosed in qualitative research manuscripts and can help inform potential bias in interpreting themes. Therefore, we included it initially.
However, Nutrients does include a short biography of the authors. And, as you kindly and correctly pointed out, there is no technical requirement for this section. Therefore, we removed it. Instead, we provided details on the potential for bias based on identities and experience in the discussion section. It reads:
“None of the authors have experienced a lack of nutrition security, nor did they have children in the NYC public school system at the time of this study. Therefore, there may be nuances of the lived experience which may not have been plainly identified with these perspectives. However, several of the authors are working parents and have had children who attended public school. In addition, the authors have spent years studying school meals and nutrition security issues, including those in NYC, and the authors have diverse ethnic identities and social experiences. Therefore, this cumulative experience may be a study strength.”
11.) Information regarding statistical analysis is completely missing. How did you calculate p-value to compare survey and focus groups? It should be clarified.
Thank you for pointing out this important omission. We have now included this section.
“2.3 Statistical Analysis
We conducted an analysis of participants who completed the initial Qualtrics survey, with demographic and other information, as well as participated in a focus group. We tested for normality and the data was found to be non-parametric. Due to a lack of normality and small sample size, for categorical comparisons of two groups (gender), we used a Fisher’s exact test comparison, while we used Kruskal-Wallis equality-of-populations rank tests for three or more groups (all other categories – age, occupation, etc.). Statistical significance was set a priori at p<0.05 for all determinations. We used Stata version 17 software to perform all statistical analyses (StataCorp LP, College Station, TX, USA).”
Results
12.) Please complete specific p-values in Table 1.
Thank you for pointing this out. Per the combined suggestions of the reviewers, we have included all p-values as values, highlighting those which were significant.
13.) While describing the Tables and headings for the Tables it should be marked that it was during COVID-19. Otherwise it may be unclear for the readers.
Thank you for pointing this out. All of the tables, except for the first which shows demographics only, have been edited to include the phrase ‘during COVID-19.’
14.) The description of each table should be placed above the table, not below.
Thank you for bringing this to our attention. We mistakenly thought that the table should be inserted after the first mention to save space, but realized the actual table should be placed after the detailed description. All tables have been placed after the descriptive text throughout the manuscript. Text was also condensed slightly, where possible.
15.) Please expand the headings for Table 3, 4, 5, 6, 7, 8, 9, 10, 11 because right now they are a bit vague.
Thank you for bringing this to our attention. We tried to make the tables more descriptive, without being unduly long, by now using the following format:
[Subtheme][Facilitators/Barrier] to grab-and-go meal participation during COVID-19.
For example:
Communication facilitators to grab-and-go meal participation during COVID-19.
16.) Line 372 – change the size and font according to the journal’s guidelines, as well as font size in Table 8.
Thank you for bringing these typographical errors to our attention. They have been corrected.
Discussion
17.) In lines 504-505 Authors must add an appropriate reference to support cited information.
We were unclear about which material needed to be cited and rewrote the paragraph and provided what we hope will be clarifying citations.
Instead of:
Providing access to school meals for children, especially during periods of emergency, is critical for the nutrition security of the most vulnerable members of society. Our results showed that governmental support of families with programs like grab-and-go meals and P-EBT are welcomed, appreciated, and impactful, but could be refined to reach more families with more immediate community feedback built into the process. OFNS, like many school meal service programs across the nation, had to undergo a herculean effort to pivot to a new, long-term feeding operation over one weekend at the start of COVID-19. Short of a formal evaluation, which would have been difficult given the emergency, receiving real-time community feedback may help in developing future programs.
It now reads:
Given that school meals have been shown to be among the healthiest meals that children consume in the U.S. and help improve academic outcomes [23], providing school meals for children, especially during periods of emergency, is critical for nutrition security. Our results showed that government support of families with programs like grab-and-go meals and P-EBT is welcomed, appreciated, and impactful, but could be refined to reach more families if there were more immediate community feedback built into the process. OFNS, like many school meal service programs across the nation, pivoted to a new grab-and-go emergency feeding program over one weekend at the start of COVID-19 [2] – undoubtedly, a herculean effort. This turned into a long-term feeding operation lasting at least through the conclusion of our research (June, 2021). Given the sudden switch to a new mode of meal service operation, an evaluation with a pre-post design was not possible. However, evaluation designs that collect real-time community feedback may help during future emergencies.
We hope these changes address any and all concerns.
Sincerely yours,
Cadenhead et al.
Round 2
Reviewer 2 Report
Authors have comprehensively responded to all my suggestions and comments and I have no further remarks.